# PagFormer: Polar Accumulator Grid Integrated into Transformers for Medical Image Segmentation

## Abstract

Recent transformers have made remarkable strides in medical image analysis, enhancing the efficacy of various downstream applications. Yet, the rich geometric patterns present in medical images offer untapped potential for further refinement. In this paper, introduce the Polar Accumulator Grid (PAGrid) and seamlessly integrate it into the transformer network, PagFormer, with an aim to improve segmentation performance for elliptical or oval objects in medical images. Inspired by both the bilateral grid, renowned for its edge-preserving filtering, and the directed accumulator, skilled at integrating geometric shapes into neural networks, PAGrid facilitates geometric-preserving filtering through a symmetric sequence of accumulating, processing, and slicing. PAGrid preserves elliptical geometry information and promotes the aggregation of global information. The symmetry between accumulation and slicing in PAGrid allows us to transition from the classic encoder-decoder architecture to an encoder-slicer design, emboddied in the PagFormer. Additionally, PAGrid's parallelization is managed with CUDA programming, and the back-propagation is enabled for neural network training. An empirical experiment on three medical image segmentation datasets — specifically, ISIC2017 and ISIC2018 datasets for skin lesions, ACDC datasets for cardiac organs, all of which contains elliptically distributed objects — reveals that our method outperforms other state-of-the-art transformers.

## 1 Introduction

Convolutional Neural Networks (ConvNets) He et al. (2016); Huang et al. (2017) and transformers Vaswani et al. (2017); Dosovitskiy et al. (2020); Liu et al. (2021) have recently achieved considerable success in diverse medical image analysis tasks. These range from the reconstruction of magnetic resonance imaging (MRI) Zhang et al. (2023c) and computed tomography (CT) Genzel et al. (2022) to lesion segmentation Zhang et al. (2021a); Rahman & Marculescu (2023a), MRI image registration Balakrishnan et al. (2019); Chen et al. (2022), and disease recognition Zhang et al. (2022). Of particular note, transformers, powered by the self-attention mechanism, consistently outpace traditional ConvNets. The performance enhancement is often attributed to the availability of large-scale datasets Ding et al. (2022); Sun et al. (2017) and the expansive effective receptive field Ding et al. (2021; 2022); Luo et al. (2016).

Efforts to integrate transformers for medical image segmentation have been substantial, as highlighted by numerous studies Rahman & Marculescu (2023a); Cao et al. (2022); Wang et al. (2022b); Bo et al. (2023); Chen et al. (2021); Rahman & Marculescu (2023b); Zhang et al. (2021c); Wang et al. (2022a). These methodologies commonly adopt two main strategies to enhance segmentation outcomes. One involves crafting specific decoders using either convolutional or transformer modules. The other integrates ConvNets with transformers using a layered structure. Still, there are moments when these configurations may miss out on specific details inherent to the task at hand, suggesting room for further refinement. The potential is particularly evident in medical image analysis tasks, especially when discerning geometric patterns of target objects in situations with limited data. This point is underscored by tasks such as identifying primary structures like the left/right ventricles in cine-MRI cardiac images or segmenting skin lesions in dermoscopy scans.

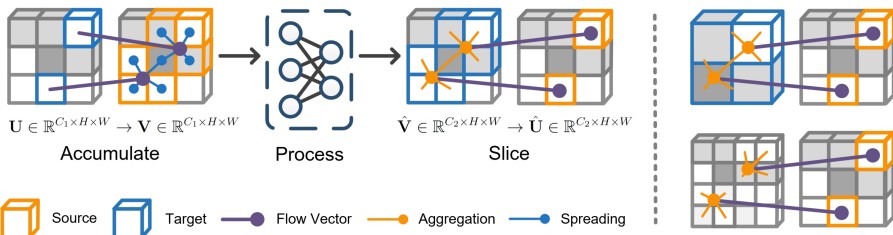

Figure 1: Overview of the proposed PAGrid. The left panel illustrates the PAGrid's accumulate-process-slice sequence for image processing. The accumulation phase employs a directed accumulator Zhang et al. (2023a) to transform input $\mathbf{U} \in \mathbb{R}^{C_1 \times H \times W}$ into $\mathbf{V} \in \mathbb{R}^{C_1 \times H \times W}$, while the slicing phase utilizes grid sampling Jaderberg et al. (2015) to convert processed $\hat{\mathbf{V}} \in \mathbb{R}^{C_2 \times H \times W}$ back to $\hat{\mathbf{U}} \in \mathbb{R}^{C_2 \times H \times W}$. The process step involves any image processing operator, with a transformer backbone network being used in our implementation. Importantly, both accumulation and slicing phases leverage a shared sampling grid, matching the shape of $\mathbf{U}$, which enables direct slicing from feature maps of varying sizes, as depicted in the right panel.

In clinical settings, numerous primary objects tend to have elliptical or oval shapes. Given their prevalence, precise delineation of these structures is advantageous for pre-treatment diagnoses and treatment planning. Therefore, in this study, we propose the Polar Accumulator Grid (PAGrid) and seamlessly integrate it into the transformer framework, PagFormer, with a specific aim to boost the segmentation precision for elliptical or oval objects in medical images. The PAGrid concept draws from the principles of both the bilateral grid Paris & Durand (2006); Chen et al. (2007) and the directed accumulator Zhang et al. (2023a), facilitating geometric-preserving processing within transformer. While the traditional bilateral grid filtering follows a splat-blur-slice sequence in image processing, our PAGrid approach adopts an accumulate-process-slice progression. Here, we favor 'accumulate' over 'splat' in the initial phase, as in image segmentation, it's more intuitive to aggregate salient features or evidence, rather than dispersing them.

The PAGrid offers two advantageous characteristics for the segmentation of elliptical objects. Firstly, unlike the traditional polar transformation using grid sampling (termed PS or polar sampling) that "pulls" a value from the source feature map for each target cell, PAGrid uses a directed accumulation strategy (termed PA or polar accumulation) Zhang et al. (2023a). In this method, every cell in the source feature map "pushes" its value to a designated cell in the target map. While the polar sampling (PS) technique can lead to information loss if the mapping between source and target isn't one-to-one, PAGrid preserves more details by pushing values from every cell in the source map.

Secondly, unlike PS and inverse PS which requires two different sampling grids for the forward and reverse processes, PAGrid simplifies this by using a single sampling grid for both accumulation and slicing, leveraging the inherent symmetry between the "push" and "pull" actions. This simplified method is particularly beneficial when embedding PAGrid within transformers. Instead of relying on traditional encoder-decoder architectures, we can directly derive the segmentation map by slicing from the intermediate feature maps generated by the backbone. This innovation leads us to adopt an encoder-slicer design.

In this study, we assess the performance of PAGrid and PagFormer across three medical image segmentation tasks: skin lesion segmentation using the ISIC2017 Codella et al. (2018) and ISIC2018 datasets Codella et al. (2019); Tschandl et al. (2018), and cine-MRI cardiac image segmentation with the ACDC dataset Bernard et al. (2018). Our contributions are threefold:

- We present the first seamless integration of a symmetric tri-phase (namely, accumulate-process-slice) image processing sequence PAGrid into contemporary neural network frameworks.

- We introduce an innovative encoder-slicer architecture, PagFormer, tailored for tasks presenting elliptical objects, as an alternative to the long-standing encoder-decoder design.

- The proposed PagFormer not only exhibits faster convergence but also surpasses the best-performing methods on the three medical image datasets.

## 2 RELATED WORKS

Dosovitskiy et al. Dosovitskiy et al. (2020) introduce the concept of vision transformers (ViTs), which rely on the self-attention mechanism and introduce fewer inductive biases compared to ConvNets. Their effectiveness is further amplified by leveraging large-scale datasets and increased model capacities. Building on this, the Swin transformer Liu et al. (2021) incorporates a shifted window-based attention strategy and features a hierarchical structure similar to ConvNets. Following this trend, several models emerge that integrate features from both ViTs and Swin transformers for medical image segmentation. For instance, the Pyramid Vision Transformer (PVT) Wang et al. (2021) combines the strengths of ConvNets and ViTs, establishing itself as a versatile backbone for dense predictions without relying on convolutions. Similarly, SwinUnet Cao et al. (2022) is a fully transformer-based model, where both the encoder and decoder segments utilize Swin transformer blocks, forming a U-shaped architecture. In contrast, TransUnet Chen et al. (2021) adopts a hybrid structure, combining ConvNet-ViT elements and integrating a U-Net-like decoder Ronneberger et al. (2015). Several other models emphasize modifications to the decoder. For example, PolypPVT Dong et al. (2021) incorporates additional attention modules, specifically CBAM Woo et al. (2018), into its decoders. Similarly, CASCADE Rahman & Marculescu (2023b) introduces a cascaded attention mechanism for the decoder.

### 2.1 LEARNING WITH GEOMETRIC PRIORS

Training deep neural networks often hinges on access to large-scale datasets Deng et al. (2009); Lin et al. (2014); Kirillov et al. (2023), which can be problematic for specific medical applications with limited data availability. For example, even when trained on extensive datasets, large vision models like SAM Kirillov et al. (2023) still lag behind specialized models in various medical imaging tasks, despite attempts at fine-tuning Ma & Wang (2023). In contrast, leveraging geometric priors can yield better results. For instance, methods such as distance transformation mapping Ma et al. (2020) and spatial information encoding Liu et al. (2018) have paved the way for edge-aware loss functions Kervadec et al. (2019); Zhang et al. (2021b); Karimi & Salcudean (2019), the inclusion of anatomical coordinates in network layers as priors Zhang et al. (2021a), and the creation of spatially covariant network weights Zhang et al. (2023b). Polar or log-polar features are widely used in various tasks, including modulation classification Teng et al. (2020), medical image segmentation Benčević et al. (2021), rotation- and scale-equivariant polar transformer networks Esteves et al. (2018), object detection Xie et al. (2020); Xu et al. (2019); Park et al. (2022), correspondence matching Ebel et al. (2019), and both cell detection Schmidt et al. (2018) and segmentation Stringer et al. (2021). Additionally, using concentric circles to model layout patterns aids in lithography hotspot detection Zhang et al. (2016; 2017) and optical proximity correction Jiang et al. (2019).

## 3 METHODOLOGY

In this section, we detail the formulation of the Polar Accumulator Grid (PAGrid) and the PagFormer. We initiate the discussion with foundational concepts from directed accumulator Zhang et al. (2023a) and grid sampling Jaderberg et al. (2015). Subsequently, we describe every step in the PAGrid processing sequence and demonstrate how it merges with the current transformer architecture. Lastly, we provide a discussion on the complexity and contrast between polar accumulator (PA) and polar sampling (PS).

### 3.1 PRELIMINARIES

Directed accumulator and grid sampling are techniques designed for differentiable image transformation within neural networks. While grid sampling is effective in various scenarios, it faces challenges with transformations that involve summing or integrating multiple values from the source feature map. This includes transformations such as the Radon transform Deans (2007), Hough transform Ballard (1981); Illingworth & Kittler (1987), rim transform Zhang et al. (2023a), and symmetric radial transform Loy & Zelinsky (2002).

We begin by discussing the fundamental equations associated with directed accumulator and grid sampling. Given a source feature map $\mathbf{U} \in \mathbb{R}^{C \times H \times W}$, and a sampling grid $\mathbf{G} \in \mathbb{R}^{2 \times H \times W} =$

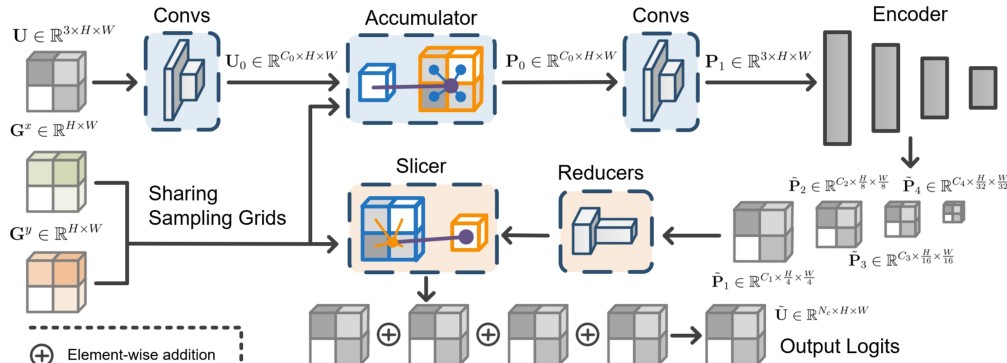

Figure 2: Schematic of the proposed encoder-slicer architecture. This structure is composed of three primary sections: the accumulator (highlighted with a light blue background), the encoder, and the slicer (indicated by the light orange background).

$(\mathbf{G}^x[k], \mathbf{G}^y[k])$, alongside a kernel function $\mathcal{K}()$, we can represent the output value of a specific cell $(i, j)$ in the target feature map $\mathbf{V} \in \mathbb{R}^{C \times H' \times W'}$ as:

$$\mathbf{V}_{ij}^c = \sum_n^H \sum_m^W \mathbf{U}_{nm}^c \mathcal{K}(\mathbf{G}_{nm}^x, i) \mathcal{K}(\mathbf{G}_{nm}^y, j), \tag{1}$$

where the kernel function $\mathcal{K}()$ can be replaced with any other specified kernels, e.g. integer sampling kernel $\delta(\lfloor \mathbf{G}_{nm}^x + 0.5 \rfloor - i) \cdot \delta(\lfloor \mathbf{G}_{nm}^y + 0.5 \rfloor - j)$ and bilinear sampling kernel $\max(0, 1 - |\mathbf{G}_{nm}^x - i|) \cdot \max(0, 1 - |\mathbf{G}_{nm}^y - j|)$. Here $\lfloor x + 0.5 \rfloor$ rounds $x$ to the nearest integer and $\delta()$ is the Kronecker delta function. The equation 1 can be denoted as a tensor mapping, $\mathcal{D}(\mathbf{U}; \mathcal{G}, \mathcal{K}) : \mathbb{R}^{C \times H \times W} \to \mathbb{R}^{C \times H' \times W'}$.

### 3.2 TRANSFORMER-BASED MEDICAL IMAGE SEGMENTATION

For grid sampling, given a source feature map $\tilde{\mathbf{V}} \in \mathbb{R}^{C \times H' \times W'}$, and considering the same variables as described in equation 1, the output value for a specific cell $(i, j)$ in the target feature map $\tilde{\mathbf{U}} \in \mathbb{R}^{C \times H \times W}$ is expressed as:

$$\tilde{\mathbf{U}}_{ij}^c = \sum_n^{H'} \sum_m^{W'} \tilde{\mathbf{V}}_{nm}^c \mathcal{K}(\mathbf{G}_{ij}^x, n) \mathcal{K}(\mathbf{G}_{ij}^y, m). \tag{2}$$

Similarly, we use notation $\mathcal{S}((\tilde{\mathbf{V}}; \mathcal{G}, \mathcal{K}) : \mathbb{R}^{C \times H' \times W'} \to \mathbb{R}^{C \times H \times W})$ to represent equation 2. Upon examining equation 1 and equation 2, we can observe that the subtle differences arise from the subscript of $\mathbf{G}$ and the input location to the kernel. In the directed accumulator, for each cell $(i, j)$ in the target feature map, values from all cells in the source feature map that point to $(i, j)$ are integrated. In contrast, with grid sampling, for each cell $(i, j)$ in the target feature map, a specific location is identified in the source feature map to retrieve the value. A vivid analogy is that directed accumulation and grid sampling operate like "push" and "pull" mechanisms on feature maps. Directed accumulation "pushes" values from the source to the target, while grid sampling "pulls" values from the source to the target.

### 3.3 PAGRID

The PAGrid adopts an accumulate-process-slice sequence for image processing, as visually illustrated in Fig. 1. In general, the PAGrid operates in three stages to process intermediate feature maps within neural networks. Initially, a polar grid is constructed from a feature map using the polar accumulator. Subsequently, processing occurs within the grid using neural networks. Lastly, the polar grid is sliced to reconstruct the output feature map. Accumulation and slicing are symmetric operations that facilitates the conversion between the input and polar grid spaces. In multi-channel feature maps, the same transformation process applies to every channel. For simplicity, throughout the remainder of this discussion, the feature map will be denoted using only its spatial dimensions.

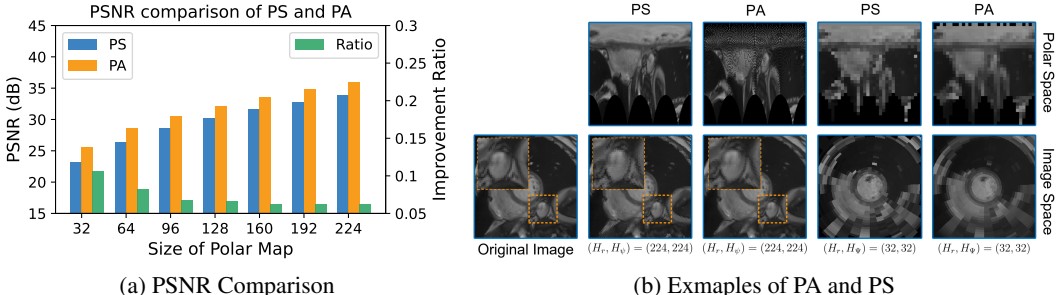

(a) PSNR Comparison

(b) Exmaples of PA and PS

Figure 3: (a) Bar plots of PSNR and (b) image reconstruction quality, comparing the performance between polar accumulator (PA) and polar sampling (PS). In (a), the ratio is calculated using the formula ratio $= \frac{\text{PSNR}_{\text{PA}} - \text{PSNR}_{\text{PS}}}{\text{PSNR}_{\text{PS}}}$. In (b), the original image is transformed into a polar space of size $(H_r, W_\psi)$ with nearest sampling kernel using either PA or PS and is then reverted back to image space. Hollowed-out regions are visible in the upper part of the PA-transformed image, contrasting with similar areas found in the peripheral regions of the PS-reconstructed image.

### 3.3.1 SAMPLING GRID

We start by defining the sampling grid necessary for creating the polar grid. With the sampling grid defined, equation 1 and equation 2 facilitate the accumulation and slicing processes. Let $\mathbf{U} \in \mathbb{R}^{H \times W}$ be the input feature map, $\mathbf{M}^x \in \mathbb{R}^{H \times W}$ (value range: $(0, H-1)$) and $\mathbf{M}^y \in \mathbb{R}^{H \times W}$ (value range: $(0, W-1)$) be the corresponding mesh grids, $(x_c, y_c)$ be the coordinate of the polar center, the value of sampling grid in the radial direction $\mathbf{G}^x$ at position $(i, j)$ can be obtained as:

$$\mathbf{G}^x_{ij} = \sqrt{(\mathbf{M}^x_{ij} - x_c)^2 + (\mathbf{M}^y_{ij} - y_c)^2}/s_r. \tag{3}$$

Here $s_r = \frac{\sqrt{H^2 + W^2}}{2 * H_r}$ represents the sampling rate in the radial direction, with $H_r$ denoting the one side of spatial dimensions of the polar grid. Similarly, the value of sampling grid in the angular direction $\mathbf{G}^y$ at position $(i, j)$ can be obtained as:

$$\mathbf{G}^y_{ij} = \arctan\left((\mathbf{M}^x_{ij} - x_c)^2 + (\mathbf{M}^y_{ij} - y_c)^2 + \pi\right)/s_\theta. \tag{4}$$

Here, $s_\theta = \frac{2\pi}{W_\psi}$ is the sampling rate in the angular direction, with $W_\psi$ denoting the other side of spatial dimensions of the polar grid. The addition of $\pi$ ensures all values fall within $(0, 2\pi)$.

### 3.3.2 HOMOGENEOUS COORDINATES

To enable geometric-preserving filtering in the PAGrid, it's important to monitor the number of pixels (or a weight) corresponding to each grid cell. Thus, during grid creation, we store homogeneous quantities $(\mathbf{V}^c_{ij} \cdot \mathbf{W}^c_{ij}, \mathbf{W}^c_{ij})$. Here, $\mathbf{W}$ can be derived from $\mathbf{W} = \mathcal{D}(\mathbf{J}; \mathcal{G}, \mathcal{K})$, where $\mathbf{J}$ is a tensor of ones. This representation simplifies the computation of weighted averages: $(w_1 v_1, w_1) + (w_2 v_2, w_2) = (w_1 v_1 + w_2 v_2, w_1 + w_2)$. Normalizing by the homogeneous coordinates $(w_1 + w_2)$ yields the anticipated averaging of $v_1$ and $v_2$, weighted by $w_1$ and $w_2$. Conceptually, the homogeneous coordinate $\mathbf{W}$ represents the importance of its associated data $\mathbf{V}$. In practice, $\mathbf{W}$ can be obtained as one of the output channels, with $\mathbf{J}$ serving as the corresponding channels of input.

### 3.3.3 POLAR GRID ACCUMULATION, PROCESSING AND SLICING

**Accumulation**: Given $\mathbf{G} = (\mathbf{G}^x, \mathbf{G}^y)$ from equation 3 and equation 4, we can generate the polar accumulator grid through equation 1 as $\mathbf{P} = \mathcal{D}(\mathbf{U}; \mathcal{G}, \mathcal{K}) \in \mathbb{R}^{H_r \times W_\psi}$, where $H_r$ and $W_\psi$ are determined by the sampling rate $s_r$ and $s_\theta$, as described in Section 3.3.1. It's worth noting that before processing $\mathbf{P}$, we use homogeneous coordinates for normalization, as described in Section 3.3.2.

**Processing**: The transformed polar grid $\mathbf{P}$ is then processed through a backbone network $f$, like the Swin transformer Liu et al. (2021). Depending on the specific network architecture, this can result in multiple outputs, expressed as $\{\tilde{\mathbf{P}}_i | i \in \{1, \dots, N_{\text{outs}}\}\}$. For instance, the Swin transformer yields four outputs, each with different spatial dimensions.

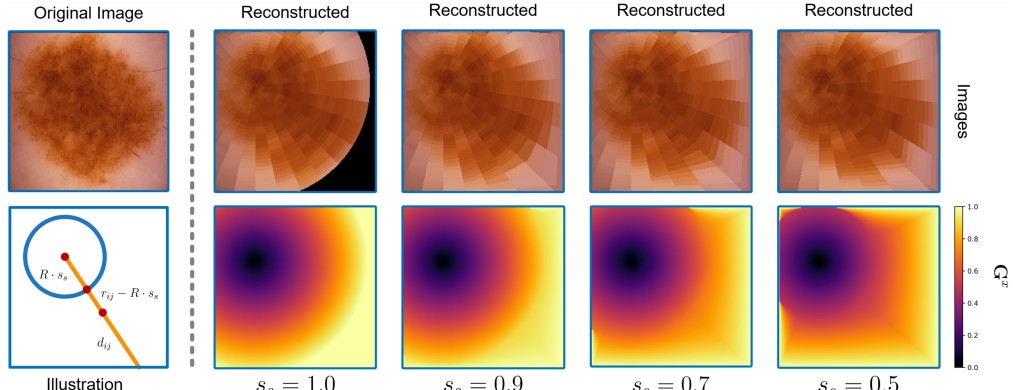

Figure 4: Visual representation of the impact of scale factors on the sampling grid $\mathbf{G}^x$ and the reconstructed image. To enhance image readability, the nearest sampling kernel is employed with a polar grid size of $(H_r = 32, W_\psi = 32)$. The sampling grid $\mathbf{G}^x$ is normalized in the range of $(0, 1)$. The left-bottom corner illustrates the equation 5.

**Slicing**: With the $i_{th}$ processed polar grid $\tilde{\mathbf{P}}_i = f(\mathbf{P})_i$ and the sampling grid $\mathbf{G}$, we apply equation equation 2 to transform the processed polar representation back into the image space, denoted as $\tilde{\mathbf{U}}_i = \mathcal{S}(\tilde{\mathbf{P}}_i; \mathcal{G}, \mathcal{K})$.

### 3.4 Integrating PAGrid into Transformers

We showcase PAGrid integration with the hierarchical Swin transformer, but it's adaptable to other encoder networks, including ConvNets He et al. (2016); Huang et al. (2017) and transformers like PVT Wang et al. (2021) and ViTs Dosovitskiy et al. (2020). As illustrated in Fig. 2, the encoder-slicer architecture comprises three main components: the accumulator, encoder, and slicer sections. The network takes as inputs the raw image $\mathbf{U} \in \mathbb{R}^{3 \times H \times W}$ and the sampling grids $\mathbf{G}^x, \mathbf{G}^y \in \mathbb{R}^{H \times W}$, with the polar grid size set to match the input image size $(H_r = H, W = W_\psi)$. The raw image is first processed through several convolutional blocks (Convs) to extract the initial feature map $\mathbf{U}_0 \in \mathbb{R}^{C_0 \times H \times W}$. These extracted features, along with the sampling grids $\mathbf{G}^x$ and $\mathbf{G}^y$, are then fed into the accumulator to transform the data into the polar grid space $\mathbf{P}_0 \in \mathbb{R}^{C_0 \times H \times W}$. After this transformation, the polar grid proceeds through additional Convs, adapting the feature map to $\mathbf{P}_1 \in \mathbb{R}^{3 \times H \times W}$, priming it for the encoder. This adjustment facilitates the effective utilization of the encoder's pre-trained weights to boost performance.

The encoder generates intermediate feature maps $\tilde{\mathbf{P}}i \in \mathbb{R}^{C_i \times H_i \times W_i} \mid 1 \le i \le Nouts$, with $N_{outs} = 4$, $H_i = \frac{H}{2^{i+1}}$, and $W_i = \frac{W}{2^{i+1}}$ specifically for the Swin transformer. Each of these feature maps is then processed through a distinct reducer, composed of linear layers, to diminish the channel dimension from $C_i$ to $N_c$. Here, $N_c$ specifies the number of categories pertinent to the task at hand. Finally, all reduced feature maps are transformed back from the polar grid space to the image space and are summed to yield the final logits $\tilde{\mathbf{U}} \in \mathbb{R}^{N_c \times H \times W}$.

### 3.5 Analysis of Characteristics and Complexity of PAGrid

The PAGrid, utilizing directed accumulator, maintains the rotation-equivariance inherent in polar transformation and also offers two additional benefits over traditional sampling-based polar methods Esteves et al. (2018); Benčević et al. (2021). These advantages include the retention of more comprehensive information from the source feature maps and enhanced flexibility in employing polar sampling grids.

**The Retention of More Comprehensive Information**: In the PS technique, each cell in the target feature map "pulls" a value from a specific cell (four cells using bilinear kernel) in the source feature map. This mechanism can result in potential information loss, especially when the mapping from source to target is not one-to-one. On the other hand, the PA approach ensures that each cell in the source feature map "pushes" its value to a cell (four cells using bilinear kernel) in the target feature map. Even though the values are smoothed during the normalization of homogeneous coordinates,

Table 1: A comparative analysis of the performance between the proposed PagFormer and other established methods on the ISIC2017 and ISIC2018 datasets. The best-performing metric is bolded.

| Model | ISIC 2017 Avg. Dice (%) | ISIC 2018 Avg. Dice (%) |
|---|---|---|
| U-Net Ronneberger et al. (2015) | 81.96 | 84.34 |
| AttUnet Oktay et al. (2018) | 81.68 | 85.84 |
| SAM Ma & Wang (2023) | 81.22 | 87.1 |
| TransUnet Chen et al. (2021) | 83.85 | 88.93 |
| SwinUnet Cao et al. (2022) | 83.69 | 88.96 |
| PolypPVT Bo et al. (2023) | 84.57 | 88.39 |
| PVT-Cascade Rahman & Marculescu (2023a) | 84.06 | 88.51 |
| PagFormer (Ours) | **85.28** | **89.70** |

the PA method ensures that every piece of information from the source is considered, reducing the risk of information loss.

## 3.6 Transformer-based Medical Image Segmentation

We first use nearest sampling kernel to show that PA-reconstructed images have visually higher quality than PS-reconstructed images, as shown in Fig. 3b. We then conducted a quantitative experiment to validate this claim, utilizing all the testing images from the ISIC2018 dataset, each resized to a $224 \times 224$ resolution. In the PA method, we sequentially applied the polar accumulator and slicer without an intervening processing phase. For the PS approach, polar sampling and inverse polar sampling were applied in sequence. The quality of the reconstructions was evaluated using the Peak Signal-to-Noise Ratio (PSNR), and an average score was calculated to compare the fidelity of images reconstructed by PA and PS to the original images. As illustrated in Fig. 3a, PA consistently outperforms PS in terms of image reconstruction quality. This improvement is more pronounced when the polar grid size is reduced from 224 to 32, showcasing the effectiveness of PA in preserving image details even at lower resolutions.

**The Enhanced Flexibility in Employing Polar Sampling Grids**: Similar to the bilateral grid Chen et al. (2007), the accumulation and slicing operations in PAGrid are symmetric, allowing the use of a single sampling grid for both forward and inverse polar transformations. This symmetry simplifies the architecture and processing steps, paving the way for the introduction of a novel encoder-slicer design. This new approach negates the need for complex decoders, a common requirement in previous methods such as Bo et al. (2023); Chen et al. (2021); Rahman & Marculescu (2023a), streamlining the process and potentially enhancing performance and efficiency.

Additionally, this symmetric characteristic simplifies the adjustment of sampling grids, promoting their easy integration into neural networks. In scenarios where objects are not centrally located in images, it is typically necessary to identify the object's center before executing the transformation. Utilizing the conventional PS method can lead to issues, such as parts of the reconstructed image being lost, as depicted in the second column in Fig. 4. To address the issue of lost parts, we can modify equation 3. Let $s_s$ be a new variable introduced as a scale factor to constrain the sampling grid both inside and outside the core circle. The sampling grid $\mathbf{G}_x$ can be redesigned as follows:

$$\begin{cases} \mathbf{G}_{ij}^x = \frac{s_s \cdot r_{ij}}{s_r}, & \text{if } r_{ij} \leq R \cdot s_s, \\ \mathbf{G}_{ij}^x = \frac{(1-s_s)(r_{ij} - R \cdot s_s)}{d_{ij} \cdot s_r} + \frac{s_s}{s_r}, & \text{Others}, \end{cases} \tag{5}$$

where $R = \frac{\sqrt{H^2 + W^2}}{2}$ represents half of the diagonal length, $r_{ij} = \sqrt{(\mathbf{M}_{ij}^x - x_c)^2 + (\mathbf{M}_{ij}^y - y_c)^2}$ is the distance from the pixel to the center of the image, and $d_{ij}$ is the distance to the image boundary along the line passing through $(x_c, y_c)$, the polar center. When $s_s = 1$, equation 5 simplifies to equation 3. Visual examples of the effects of varying $s_s$ and visual illustration of variables used in equation 5 are provided in Fig. 4.

Table 2: A comparative analysis highlighting the performance of our proposed PagFormer alongside other notable methods on the ACDC dataset is presented. Dice scores for RV, LV, and Myo are reported individually, and an average score is also provided. The highest performing metric in each category is highlighted in bold for easy reference.

| Model | RV (%) | Myo (%) | LV (%) | Avg. Dice (%) |
|---|---|---|---|---|
| U-Net Ronneberger et al. (2015) | 87.10 | 80.63 | 94.92 | 87.55 |
| AttUnet Oktay et al. (2018) | 87.58 | 79.20 | 93.47 | 86.75 |
| SAM Ma & Wang (2023) | 74.92 | 76.09 | 88.51 | 79.83 |
| TransUnet Chen et al. (2021) | 86.67 | 87.27 | 95.18 | 89.71 |
| SwinUnet Cao et al. (2022) | 88.89 | 87.98 | 95.31 | 90.73 |
| PolypPVT Bo et al. (2023) | 87.95 | 87.83 | 95.20 | 90.33 |
| PVT-Cascade Rahman & Marculescu (2023a) | 89.72 | 88.59 | 95.18 | 91.16 |
| PagFormer (Ours) | **90.39** | **89.90** | **95.59** | **91.96** |

## 4 EXPERIMENTS AND RESULTS

In this section, we start by benchmarking PagFormer against other leading methods to highlight its effectiveness. Following that, we explore the impact of changing scale factors and the number of polar grids derived from the backbone network.

### 4.1 DATASETS AND EVALUATION METRICS

**ACDC Dataset**: The ACDC dataset Bernard et al. (2018) is comprised of 100 cine-MRI cardiac scans gathered from a diverse group of patients. Each scan reveals three distinct organs: the right ventricle (RV), left ventricle (LV), and the myocardium (Myo). We adhere to the evaluation protocol established in previous research Rahman & Marculescu (2023a), allocating 70 cases (equivalent to 1930 axial slices) for training, 10 for validation, and the remaining 20 for testing purposes. **ISIC2017 Dataset**: The ISIC2017 dataset Codella et al. (2018) is comprised of 2000 training images, 150 validation images, and 600 test images. Contrary to the approaches of prior studies, we opt to use the dataset partitions for training, validation, and testing as provided on the official website [1], avoiding manual splitting. **ISIC2018 Dataset**: The ISIC2018 dataset Codella et al. (2019); Tschandl et al. (2018) consists of 2594 training images, 100 validation images, and 1000 test images. In contrast to previous studies, we adhere to the dataset partitions for training, validation, and testing that are officially provided on the dataset's website[2], eliminating the need for manual splitting. **Evaluation Metrics**: We adopt the Dice score as the evaluation metric for all three datasets. Specifically, for the ACDC dataset, we report the Dice scores for the RV, LV, Myo, and their average.

### 4.2 COMPARATORS AND IMPLEMENTATION DETAILS

We benchmark our proposed method against other prominent models including SAM Ma & Wang (2023), TransUnet Chen et al. (2021), SwinUnet Cao et al. (2022), PolypPVT Bo et al. (2023), PVT-Cascade Rahman & Marculescu (2023a), U-Net Ronneberger et al. (2015), and Attention-Unet (AttUnet) Oktay et al. (2018). For a fair comparison, we incorporate U-Net, AttUnet and SAM into our framework. For the remaining models, we replicate their results using their own implementations to ensure consistency and accuracy in the comparative analysis.

**Implementation Details**: All experiments were executed using Python 3.7, with network models developed utilizing PyTorch library Paszke et al. (2019) version 1.9.0. The training was carried out on a machine powered by an A100 GPU. The directed accumulator was crafted and executed in CUDA version 11.1, while for slicing, the built-in *grid_sample()* function in PyTorch was employed.

Optimization was performed using the Adam optimizer Kingma & Ba (2014), initiated with a learning rate of 1e-4. We employed a multi-step learning rate scheduler that reduced the learning rate by half at 50%, 70%, and 90% of the total epochs. All images were adjusted to a resolution of $(224, 224)$ and training was conducted with a mini-batch size of 12. The training process was con-

---

[1]https://challenge.isic-archive.com/data/#2017

[2]https://challenge.isic-archive.com/data/#2018

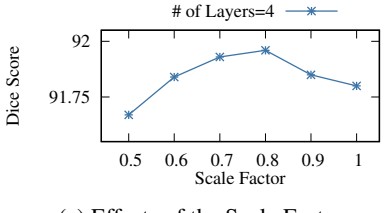 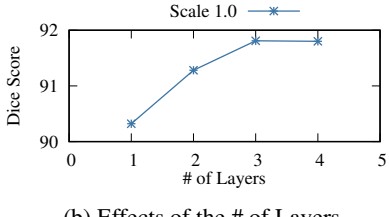

(a) Effects of the Scale Factor          (b) Effects of the # of Layers

Figure 5: Ablation study on the effects of the scale factor (a) and the # of intermediate feature maps used (b) for ACDC dataset.

cluded after 100 epochs. For our comparisons, though PAGrid is adaptable to various backbone networks, we have chosen to integrate it with the base Swin Transformer to form the PagFormer. We adopt the approach from previous research Benčević et al. (2021), employing SwinUnet to train a heat map generator for locating the center of objects. In the case of the ACDC dataset, we specifically focus on identifying the center of the left ventricle.

## 4.3 QUANTITATIVE RESULTS

Table 1 presents a comparison of average Dice scores (%) between the proposed PagFormer and other state-of-the-art models on the ISIC 2017 and ISIC 2018 datasets. PagFormer outperforms all other listed models, achieving the highest average Dice scores of 85.28% and 89.70% on ISIC 2017 and ISIC 2018, respectively. Compared to the next best model, PolypPVT for ISIC 2017 and SwinUnet for ISIC 2018, PagFormer shows an improvement rate of approximately 0.84% and 0.83%, respectively.

Table 2 showcases the performance comparison on the ACDC dataset of various models, including the proposed PagFormer. The models are evaluated based on the Dice score (%) for three different organs—RV, Myo, and LV—as well as the average Dice score across all three. PagFormer excels in all categories, registering Dice scores of 90.39%, 89.90%, and 95.59% for RV, Myo, and LV, respectively, leading to the highest average Dice score of 91.96%. When compared to the second-best model, PVT-Cascade, PagFormer exhibits an improvement of about 0.88% in the average Dice score, underscoring its superior performance in cardiac MRI segmentation tasks.

## 4.4 ABLATION ANALYSIS

We conducted an ablation study on the ACDC dataset, focusing on the impact of scale factors and the number of intermediate feature maps employed. **Effects of the Scale Factor**: As illustrated in Fig. 5b, we maintained the number of intermediate feature maps at four and varied the scale factors. The performance rose starting from $s_s = 0.5$, peaked at $s_s = 0.8$, and then declined. Interestingly, for the ISIC datasets, the peak performance was observed at $s_s = 0.9$. This variation indicates that the average object sizes in the ISIC datasets are comparatively larger than those in the ACDC dataset. **Effects of the # of Feature Maps**: Fig. 5b also provides insights into the performance variation with different numbers of feature maps, while keeping the scale factor $s_s = 1.0$ constant. The data suggests a direct correlation between the number of intermediate feature maps used and the performance improvement. A notable observation is that employing even a single feature map of size $(7, 7)$ enables our model to surpass the performance of all competing methods, with the exception of PVT-Cascade.

## 5 CONCLUSIONS

In this study, we introduced PagFormer, a model integrating Polar Accumulator Grid (PAGrid) with transformer architectures for improved medical image segmentation. PAGrid ensures efficient image transformation and processing, overcoming limitations of traditional methods. Our experiments on ISIC2017, ISIC2018, and ACDC datasets confirmed PagFormer's superior performance and effectiveness. The model's adaptability and efficiency were validated through ablation studies, underscoring its potential for advanced biomedical image processing applications.

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
