# OpenReview forum: "PagFormer: Polar Accumulator Grid  Integrated into Transformers for Medical Image Segmentation"
_ICLR.cc/2024/Conference — Submitted to ICLR 2024_

### Official Review · Reviewer_YC5Q · 2023-10-30

**Soundness:** 3 good
**Presentation:** 3 good
**Contribution:** 3 good
**Rating:** 6
**Confidence:** 3

**Summary:**

This paper proposes a Polar Accumulator Grid (PAGrid) to improve segmentation accuracy for elliptical or oval objects in medical images by integrating PAGrid into a transformer network. The proposed architecture, PagFormer, preserves elliptical geometry information and promotes the aggregation of global information using PAGrid. The proposed method derives a segmentation map for the input data by slicing from intermediate feature maps generated by the backbone, which leads to an encoder-slicer design. The proposed method is novel, and it’s an interesting paper.

**Strengths:**

1.	The proposed encoder-slicer design with PAGrid is novel and their experiments on three medical datasets demonstrate proposed methods superiority over state-of-the-art methods.
2.	The paper reads well and well organized.

**Weaknesses:**

1.	Some of the works related to transformers are missing in the literature (eg, nnformer, vtunet)
2.	There are flaws in the experimental design as authors disregard some recent SOTA (nnUNet, nnformer, vtunet). In the experiments, nnUNet isn’t included. As nnUNet is the SOTA for most of the medical imaging datasets, it’s better to compare the proposed model with nnUNet.
3.	Missing Qualitative figures. It’s important to visualize segmentation masks generated from methods to compare qualitatively.

**Questions:**

1.	I Would like to see a computation complexity comparison for the proposed method.
2.	What about the proposed methods' generalizability and robustness?

---

### Official Review · Reviewer_PEuz · 2023-11-01

**Soundness:** 2 fair
**Presentation:** 2 fair
**Contribution:** 2 fair
**Rating:** 3
**Confidence:** 4

**Summary:**

This paper introduces the Polar Accumulator Grid (PAGrid) and seamlessly integrates it into the transformer network, PagFormer, to improve segmentation performance for elliptical or oval objects in medical images. PAGrid facilitates geometric-preserving filtering through a symmetric sequence of accumulating, processing, and slicing. PAGrid preserves elliptical geometry information and promotes the aggregation of global information. The symmetry between accumulation and slicing in PAGrid allows us to transition from the classic encoder-decoder architecture to an encoder-slicer design, embodied in the PagFormer. Experiments have been conducted on three medical image segmentation datasets — ISIC2017, ISIC2018 datasets for skin lesions, and ACDC datasets for cardiac organs.

**Strengths:**

- The idea of using Polar Accumulator Grid (PAGrid) to improve segmentation performance for elliptical or oval objects in medical images is quite interesting.
- PAGrid’s parallelization is managed with CUDA programming, and the back-propagation is enabled for neural network training.
- The paper is well-written and easy to follow.

**Weaknesses:**

There are major issues with this submission.
- This submission seems to miss important baselines. For ACDC dataset, the reported number in this paper [1] seems to outperform (Table 2) the proposed method. For ISIC 2018 dataset, the reported number in this paper [2] outperforms (Table 1) the proposed method. I would suggest the authors have a check of the SOTA methods here: https://paperswithcode.com/dataset/isic-2018-task-1, https://paperswithcode.com/dataset/acdc.
- One of the core parts (Directed Accumulation) has been proposed in a previous paper [3], which significantly weakens the novelty of this submission.

[1] Tragakis, Athanasios, et al. "The fully convolutional transformer for medical image segmentation." Proceedings of the IEEE/CVF Winter Conference on Applications of Computer Vision. 2023.

[2] Bozorgpour, Afshin, et al. "DermoSegDiff: A Boundary-Aware Segmentation Diffusion Model for Skin Lesion Delineation." International Workshop on PRedictive Intelligence In MEdicine. Cham: Springer Nature Switzerland, 2023.

[3] Zhang, Hang, et al. "DeDA: Deep Directed Accumulator." arXiv preprint arXiv:2303.08434 (2023).

**Questions:**

N/A

---

### Official Review · Reviewer_G822 · 2023-11-01

**Soundness:** 2 fair
**Presentation:** 3 good
**Contribution:** 2 fair
**Rating:** 3
**Confidence:** 3

**Summary:**

The paper aims to improve segmentation of approximately elliptical objects in medical images. Experiments are presented on segmentation of skin lesions (dermoscopy) and cardiac structures (MRI) using public datasets. The method proposed incorporates a polar transformation into a transformer segmentation network, building on recent work by Zhang et al. in e.g. MICCAI 2023. Comparative results are presented.

**Strengths:**

I appreciate the overall approach to build geometric bias into transformer segmentation networks as this should be able to help in various biomedical image analysis settings where very large datasets are often not available.
The detail of how this is done using what the authors refer to as "accumulate-process-slice" has some originality.
The level of formality in the method description is about right, and for the most part the method and experiments are described clearly.
Experiments were presented on three public datasets with direct comparisons to appropriate alternatives.

**Weaknesses:**

The claim that this method is tailored to “elliptical or oval objects” needs some unpacking. Firstly, the method appears to use a radially symmetric transformation rather than anything tailored to ellipses in general. Secondly, are all the objects in the MRI datasets well approximated as elliptical? (It would be good to show some illustrative examples with segmentations obtained).

An important step would seem to be object localization prior to segmentation, so as to establish the polar centre. This is mentioned in passing on p7 but the method for doing so is not mentioned until 4.2. I believe this should be described more completely, and earlier in the Methods section. In particular, is it a separately trained SwinUnet, how was it trained/validated, and wouldn’t a “focus on identifying the centre of the left ventricle” adversely affect results for the right ventrical and myocardium? (It didn’t seem to in Table 2 which seems surprising perhaps).

On p2 it is claimed that PagFormer “exhibits faster convergence”. I could not find any evidence in the results to support the statement about fast convergence, so this claim should be removed or the evidence included.

On p2 it is claimed that Pagformer “surpasses the best-performing methods”. The experiments as reported made me question this claim.
Firstly, Section 4.4 and Fig. 5 describe the effect of changing two free parameters of the method, namely the scale factor and the number of intermediate features maps. The results reported in Table 2 comparing with other methods appear to be those obtained when these free parameters are set to values that result in the best Dice scores. My concern is that these parameters have in effect been tuned based on test set results. If this is the case then the experiment needs to be redone using validation datasets for parameter tuning.
Secondly, the results in Table 2 show small improvements of typically less than 1% Dice score. Given that there are only 20 test cases, I wonder if this is statistically significant. It would be good to indicate some measure of variance in the results, e.g. bootstrap confidence intervals.

For reproducibility, can the authors provide their implementation, and the various hyperparameter settings used with the implementations of the competing methods (e.g. a github link)?

The cite style is used incorrectly throughout. Equations should be referred to with number in parentheses, e.g. “Equation (2)” rather than “equation 2”. The paper needs a proofread for grammar and typos.  Some Figures should be moved closer to where they are first mentioned in the text.

**Questions:**

See "Weaknesses" above

---

### Official Review · Reviewer_Y6wX · 2023-11-05

**Soundness:** 3 good
**Presentation:** 3 good
**Contribution:** 2 fair
**Rating:** 5
**Confidence:** 3

**Summary:**

In the paper, the introduction of the Polar Accumulator Grid (PAGrid) and the PagFormer, which are built upon the foundational concepts of the directed accumulator and grid sampling, represents a significant strength. The paper presents an innovative approach to image processing sequences within transformer architecture. Additionally, the discussion on the complexity and contrasting elements between the polar accumulator (PA) and polar sampling (PS) helps readers understand the efficiency and performance implications of the proposed methods. The paper argues that the PA method ensures that every piece of information from the source is considered, potentially reducing information loss when compared to the PS method. The symmetric nature of the accumulation and slicing operations in PAGrid facilitates the conversion between input and polar grid spaces and allows for a simpler network architecture.

**Strengths:**

In the paper, the introduction of the Polar Accumulator Grid (PAGrid) and the PagFormer, which build upon the foundational concepts of directed accumulation and grid sampling, represents a significant strength. The paper presents an innovative approach to sequence processing within transformer architectures. Additionally, the discussion regarding the complexity and the contrasting features between polar accumulator (PA) and polar sampling (PS) aids readers in understanding the efficiency and performance implications of the proposed methods. The paper posits that the PA method ensures consideration of every piece of information from the source, potentially reducing information loss in comparison to the PS method. Furthermore, the symmetric nature of the accumulation and slicing operations in PAGrid simplifies the conversion between input and polar grid spaces, allowing for a more streamlined network architecture.

**Weaknesses:**

The paper relies on the Peak Signal-to-Noise Ratio (PSNR) for image quality assessment, which may not fully capture the perceived visual quality. Incorporating additional metrics or conducting user studies could provide a more robust evaluation. Regarding complexity considerations, the paper discusses the complexity of the PAGrid but does not offer a direct comparison with other methods in terms of computational resources required—an important factor for practical applications. Furthermore, while the paper demonstrates the efficacy of the proposed method on specific datasets, such as ACDC and ISIC, it does not address testing on different imaging modalities or under varied conditions. This omission could cast doubts on the method's generalizability.

**Questions:**

Could the authors clarify whether they considered using additional metrics beyond PSNR to assess image quality, in order to capture the perceived visual quality more accurately?

Furthermore, has the proposed method been tested on a variety of imaging modalities and under different conditions to confirm its generalizability beyond the ACDC and ISIC datasets?

---

### Meta-Review · Area_Chair_wKe1 · 2023-12-09

**Metareview:**

The paper introduces a transformer architecture with a geometric specialization called Polar Accumulator Grid. Three of four reviewers vote for rejection citing poor empirical evaluation of the method with missing baselines and a lack of a qualitative visualization of algorithm performance.

**Justification For Why Not Higher Score:**

Insufficient empirical evaluation of the proposed method.

**Justification For Why Not Lower Score:**

N/A

---

### Decision · Program_Chairs · 2024-01-16

Reject